# A Promising Approach to Treat Psoriasis: Inhibiting Cytochrome P450 3A4 Metabolism to Enhance Desoximetasone Therapy

**DOI:** 10.3390/pharmaceutics15082016

**Published:** 2023-07-25

**Authors:** Jiun-Wen Guo, Yu-Pin Cheng, Cherng-Jyr Lim, Chih-Yi Liu, Shiou-Hwa Jee

**Affiliations:** 1Department of Medical Research, Cathay General Hospital, Taipei 10630, Taiwan; 2Department of Dermatology, Cathay General Hospital, Taipei 10630, Taiwan; m0587304.05g@g2.nctu.edu.tw; 3Department of Emergency Medicine, Cathay General Hospital, Taipei 10630, Taiwan; chad_911@hotmail.com; 4Division of Pathology, Sijhih Cathay General Hospital, New Taipei City 22174, Taiwan; cyl1124@gmail.com; 5Department of Dermatology, College of Medicine, National Taiwan University, Taipei 10617, Taiwan; shiouhwa@gmail.com

**Keywords:** cytochrome P450 3A4, desoximetasone, psoriasis, microemulsion, topical, skin barrier

## Abstract

(1) Background: Human keratinocytes and murine skin express various cytochrome P450 enzymes. These include cytochrome P450 3A4, which may participate in the metabolism of cytochrome P450 3A4 substrate drugs. Desoximetasone, a topical corticosteroid and cytochrome P450 3A4 substrate, is used to treat skin conditions such as skin allergies, atopic dermatitis, and psoriasis. In this study, we aimed to investigate the anti-psoriatic effect of a low dose of desoximetasone by inhibiting cytochrome P450 3A4 metabolism in the epidermis. (2) Methods: Psoriasis-like skin was induced in BALB/c mice via the topical administration of imiquimod. The mice were then topically treated with 0.01–0.05% desoximetasone loaded into a cytochrome P450 3A4 enzyme inhibitor excipient base emollient microemulsion, 0.25% commercial desoximetasone ointment, or 0.5 mg/gm clobetasol ointment. (3) Results: The topical application of 0.05% desoximetasone loaded into a cytochrome P450 3A4 enzyme inhibitor excipient base emollient formulation restored the imiquimod-induced skin barrier disruption and resulted in fewer severe clinical and pathological features compared with the treatments with 0.25% commercial desoximetasone ointment and 0.5 mg/gm clobetasol ointment. (4) Conclusions: The cytochrome P450 3A4 enzyme inhibitor excipient base emollient formulation improved and prolonged the therapeutic effect of cytochrome P450 3A4 substrate drugs and may be a promising approach for psoriasis treatment.

## 1. Introduction

Cytochrome P450 (CYP) enzymes are a group of heme-containing enzymes that play a crucial role in the metabolism of xenobiotics, including drugs and environmental toxins [1]. Numerous studies have demonstrated that keratinocytes express different subfamilies of CYP enzymes, including 1A1, 1B1, 2B6, 2E1, 3A4, and 3A5 [2,3,4,5,6,7]. Desoximetasone (C_22_H_29_FO_4_; MW = 376.5), a substrate of CYP3A4, is a potent topical corticosteroid used to alleviate various skin conditions such as skin allergies, atopic dermatitis, and psoriasis [8]. Desoximetasone is also safe with topical bioavailability steroids [8]; it is used as a routine topical steroid at several medical centers in Taiwan, including Cathay General Hospital.

Microemulsions are micron- to nanometer-sized emulsion droplets of oil, water, and surfactants that are transparent and thermodynamically stable. Pharmaceutical excipients are critical in pharmaceuticals, and surfactants are the most potent inhibitors due to their ability to disturb the microenvironment of enzymes [9]. Previous studies have reported that both Tween-80 and Cremophor RH40 can inhibit CYP3A4 activity in hepatocytes and liver microsomes [10,11]. Ren and colleagues observed that surfactants such as polysorbate 20, polyoxyl 35 castor oil, polyoxyl 40 stearate, and poloxamer 188 could inhibit CYP3A and alter the bioavailability of its substrates [12]. Rao and Zhao demonstrated that Cremophor RH40, Cremophor EL, and Tween-80 could reduce the metabolism of P-glycoprotein and CYP3A4 substrates [13,14]. Kwon et al. reported that Pluronic P85 and Tween-80 could inhibit P-glycoprotein and CYP3A4 [15]. A potential theory for the substrate drugs of CYP3A was proposed in our previous study through an excipient inhibitor-based microemulsion formulation design [16]. We developed a topical bicontinuous microemulsion delivery system using Cremophor RH40 and Tween-80 for the treatment of inflammatory skin conditions, hydration of dry skin, and delivery of large-molecular-weight compounds [17,18]. The promising potential of a microemulsion formulation comprising Tween-80 and Cremophor RH40 for CYP3A4 substrates in dermatology applications is the subject of this study. Specifically, the anti-psoriatic effect of a low dose of desoximetasone loaded into a formulation was investigated. The results of this study may have significant implications for the future design of microemulsion-based drug delivery systems to treat inflammatory skin conditions.

## 2. Materials and Methods

### 2.1. Materials

Merck KGaA (Darmstadt, Germany) supplied the desoximetasone, betamethasone, and microemulsion excipients, as previously described in [18]. The other chemicals were of analytical grade.

### 2.2. Microemulsion Preparation 

The microemulsion was prepared as previously described in [18]. 

### 2.3. Animals

Male BALB/c mice (6–8 weeks old) were obtained from the National Laboratory Animal Center (Tainan, Taiwan) and housed under controlled environmental conditions (22 ± 2 °C temperature; 40% humidity; 12 h light–dark cycle). Various studies were conducted to investigate the therapeutic effects and pharmacokinetic parameters of desoximetasone, including dose selection, drug comparison, drug crossover, and pharmacokinetic analyses. During the experimental period, each mouse was separately housed in a cage and provided with toys.

#### 2.3.1. Imiquimod-Induced Psoriasis-like Animal Model of Skin

To induce psoriasis-like skin lesions, male BALB/c mice aged 6–8 weeks were used in accordance with previous methods [18]. Specifically, 5% imiquimod (IMQ) cream was topically applied to the skin of the mice.

#### 2.3.2. CYP3A4 Protein Expression Study

Twenty-four mice were randomly and equally assigned to one of four groups: (a) normal, an untreated normal group that received no treatment as a negative control; (b) control, a control group that only received IMQ induction; (c) vehicle, a vehicle group that received IMQ induction plus treatment with a microemulsion vehicle; (d) CF, a CF group that received IMQ induction plus treatment with a commercially available moisturizing cream with the same percentage of CYP3A4-inhibitory components used in the designed formulation.

#### 2.3.3. Drug Dose Selection Study 

Thirty mice were randomly and equally assigned to one of five groups: (a) normal, an untreated normal group that received no treatment as a negative control; (b) control, a group that only received IMQ induction; (c) vehicle, a group that received IMQ induction plus treatment with a microemulsion vehicle; (d) 0.01% DXM, a group that received IMQ induction plus treatment with 0.01% desoximetasone (DXM) in a microemulsion; (e) 0.05% DXM, a group that received IMQ induction plus treatment with 0.05% desoximetasone in a microemulsion. Following the application of IMQ for 3–4 h, 100 μL of a microemulsion vehicle, 0.01% desoximetasone in a microemulsion, or 0.05% desoximetasone in a microemulsion was applied once daily to the dorsal skin of the mice. 

#### 2.3.4. Drug Comparison Study 

Following the drug dose selection study, thirty mice were randomly and equally assigned to one of six groups: (a) an untreated normal group; (b) a control group; (c) a vehicle group; (d) a 0.05% DXM group; (e) an ESP group (Esperson; 0.25% desoximetasone ointment; Sanofi, Handok Inc., Seoul, Korea); (f) a CLO group (0.5 mg/g clobetasol ointment; Sinphar Pharmaceutical Co., Ltd., Taipei, Taiwan). Following the application of IMQ for 3–4 h, 100 μL of 0.05% desoximetasone in a microemulsion, 60 mg of Esperson, or 60 mg of clobetasol was applied once daily to the dorsal skin of the mice. 

#### 2.3.5. Drug Crossover Study

Twelve mice were randomly and equally assigned to one of three groups for the drug crossover study: (a) DXM to NT group, a DXM (0.05% desoximetasone) to no treatment (NT) group, where mice received IMQ-induced psoriasis-like skin followed by 6 days of DXM treatment and were then crossed over to no treatment; (b) DXM to ESP, a DXM to ESP (Esperson; 0.25% desoximetasone ointment; Sanofi, Handok Inc., Seoul, Korea) treatment group, where mice received IMQ-induced psoriasis-like skin followed by 6 days of DXM treatment and were then crossed over to the ESP treatment for 9 days; (c) ESP to DXM, an ESP to DXM treatment group, where mice received IMQ-induced psoriasis-like skin followed by 6 days of ESP treatment and were then crossed over to the DXM treatment for 9 days. Following the application of IMQ for 3–4 h, the mice received once-daily dorsal skin applications of 100 μL of 0.05% desoximetasone in a microemulsion or 60 mg of Esperson.

#### 2.3.6. Permeation Studies 

Eighteen normal untreated mice were sacrificed after anesthesia. Full-thickness dorsal skin was collected [18]. Subsequently, the skin was randomly and equally assigned to either the DXM or the ESP group.

### 2.4. Assessment of Barrier Function 

The dorsal surfaces of the mice were evaluated for transepidermal water loss (TEWL), skin hydration, and skin erythema values before the drug application (Day 0) and after the drug-loaded microemulsion application on Day 6 and Day 15 (crossover study) using an MPA 2 system equipped with Tewameter TM300, Corneometer CM825, and Mexmeter MX18 probes (Courage and Khazaka, Köln, Germany).

### 2.5. Collection of Skin Specimens

After evaluating the barrier functions, the mice were sacrificed, and their skins were divided into two samples. One sample was used for histological staining, including hematoxylin and eosin (H&E) and CYP3A4 immunohistochemical staining. The other sample was used for protein extraction.

### 2.6. Determination of Inflammatory Cytokine Proteins

Custom-made mouse LEGENDplex kits from BioLegend (San Diego, CA, USA) were used to determine cytokines such as interleukin 17A (IL-17A), interleukin 17F (IL-17F), interleukin 22 (IL-22), interleukin 23 (IL-23), and tumor necrosis factor alpha (TNF-α). The protein extraction and cytokine analysis were conducted as previously described in [18].

### 2.7. Immunohistochemical Staining for CYP3A4

Skin sections with a thickness of 5 µm were subjected to immunohistochemical staining using primary antibodies against CYP3A4 obtained from the Proteintech Group (Rosemont, IL, USA) following the detailed procedures described in a previous study [18]. 

#### Immuno-Intensity Counting 

The detailed procedures for immuno-intensity counting were described in a previous study [18]. 

### 2.8. Permeation Study

To determine the permeation parameters, in vitro skin permeation and skin deposition studies were conducted.

#### 2.8.1. In Vitro Skin Permeation Study

Skin samples with an area of 0.985 cm^2^ were mounted onto diffusion cells. The donor cells were loaded with either 200 μL of 0.05% desoximetasone or 200 mg of Esperson ointment following the detailed procedures previously described in [18]. 

#### 2.8.2. Permeation Data Analysis

The permeation parameters, including the cumulative amount (Q; μg/cm^2^), skin flux at a steady state (Js; μg/cm^2^/h), lag time (h), and permeability coefficient (Kp), were calculated as previously described in [18].

#### 2.8.3. Skin Deposition Study

To quantify the amount of desoximetasone deposited on the skin, a skin deposition study was conducted following previously described methods [18]. The extraction recovery rate of desoximetasone from whole skin was 91.5% ± 11.6% at a concentration of 0.1 μg/mL.

### 2.9. High-Performance Liquid Chromatography System

The samples were analyzed using a Primaide 1110 pump, a Primaide 1410 UV detector, and a Primaide 1210 autosampler (Hitachi, Tokyo, Japan). The mobile phase, consisting of methanol–water (70:30; *v*/*v*; pH 2.5–3, adjusted with orthophosphoric acid), was filtered through a 0.45 μm Millipore filter and degassed prior to use. A Mightysil RP-18 column (4.6 × 250 mm; 5 μm; Kanto Chemical Co., Tokyo, Japan) was used at a flow rate of 1 mL/min. The detection was performed at a wavelength of 245 nm at room temperature, and the sample injection volume was 20 μL [19]. The calibration curves were linear over the range of 0.05–10 μg/mL. The limit of detection was 0.01 μg/mL.

### 2.10. Safety Evaluation Study

To assess the safety of the selected formulation loaded with desoximetasone, a mouse skin irritation study and a blood desoximetasone concentration detection study were conducted.

#### 2.10.1. Skin Irritation Study 

Eight mice were randomly and equally assigned to either the untreated normal group or the DXM group (application of the selected microemulsion formulation loaded with 0.05% desoximetasone). After 6 days of application, the mice were evaluated for their barrier function and were then sacrificed to collect skin samples for histological staining. 

#### 2.10.2. Blood Desoximetasone Concentration Detection Study

To verify that the amount of topically administered desoximetasone had penetrated the systemic circulation, blood samples were collected from the hearts of the mice at the end of the skin irritation study for the detection of desoximetasone. Serum samples were separated by centrifugation, followed by the addition of equal volumes of acetonitrile to denature the proteins [16]. Subsequently, 20 μL of the resulting supernatant from the serum samples was subjected to an HPLC analysis.

### 2.11. Statistical Analysis

The data were expressed as the mean ± standard deviation (SD). The statistical significance was determined using Student’s *t*-test (Sigmaplot 10.0) for the skin permeation and deposition studies. For the other studies, statistical analysis was performed using IBM SPSS 20 software. This included a one-way ANOVA followed by a Scheffe post hoc test. A *p*-value of <0.05 was considered to be statistically significant. All statistical figures were created using Sigmaplot 10.0 software and were presented as the mean ± SD.

## 3. Results

### 3.1. The Designed Formulation Exhibited a Significant Inhibition of CYP3A4 Protein Expression

To obtain a deeper understanding of the ability of the formulated product to inhibit CYP3A4 protein expression, equivalent amounts of CYP3A4 inhibitors present in the designed formulation were added to a commercially available moisturizing cream (CF). An IMQ-induced skin lesion model was chosen, as it induced epidermal hyperproliferation and provided suitable epidermal samples for the immuno-intensity analysis. Immunohistochemical staining demonstrated a significant inhibitory effect on CYP3A4 protein expression in the mouse epidermis compared with all other groups (Figure 1A–D). The quantification of immuno-intensity AI counts from the CYP3A4 immunohistochemical staining also revealed that the formulated product exhibited a superior inhibitory effect compared with the commercially available moisturizing cream (CF) containing the same percentage of CYP3A4 inhibitors (Figure 1E; *p* < 0.01). The designed formulation displayed a substantial ability to inhibit CYP3A4 protein expression in the mouse epidermis, suggesting its potential as a topical delivery system capable of inhibiting the metabolism of CYP3A4 substrates.

### 3.2. The 0.05% DXM Formulation Was Selected as the Optimal Dose for Further Studies

Following the confirmation of the ability of the formulated product to inhibit CYP3A4 protein expression, two doses (0.01% and 0.05%) of desoximetasone (DXM)-loaded formulations were developed for an optimal dose selection study. The morphological and histopathological observations (Figure 2A,B) revealed that the 0.05% DXM treatment group exhibited a greater number of miniature skin scales and a thinner epidermis compared with the 0.01% DXM treatment group. The results obtained from the TEWL, skin hydration, and skin erythema analyses demonstrated that both 0.01% and 0.05% DXM formulations significantly restored the barrier function compared with the control and vehicle control groups (all *p* < 0.01; Figure 2C–E). Therefore, the 0.05% DXM formulation was selected for further comparisons in the therapeutic and crossover studies.

### 3.3. DXM Was More Effective Than Esperson and Clobetasol

In the psoriasis-like skin lesion model, the therapeutic efficacy of DXM was compared with ESP (Esperson) and CLO (clobetasol). The DXM treatment group exhibited miniature skin scales and a thinner epidermis similar to the normal untreated mice (Figure 3A,B), indicating its effective therapeutic effect. The mice treated with DXM demonstrated significantly lower TEWL and skin erythema values and higher skin hydration values compared with the ESP and CLO treatment groups and the control group (all *p* < 0.01; Figure 3C–E). Conversely, the CLO-treated mice showed miniature skin scales and a thinner epidermis similar to the normal untreated group on Day 12 (Appendix A). Thus, the results suggested that the developed formulation of DXM was a more effective and rapid treatment option for psoriasis-like skin lesions than Esperson and clobetasol.

### 3.4. DXM, ESP, and CLO Inhibited IL-23/IL-17/TNF-α Axis Protein Expression in Psoriasis-like Skin

The protein expression levels of IL-23, IL-17A, IL-17F, IL-22, and TNF-α were assessed using a cytokine array. The results demonstrated a significant increase in cytokine expression in the psoriasis-like skin compared with the normal untreated skin (negative control group). Treatments with DXM, ESP, and CLO led to the inhibition of the protein expression in the IL-23/IL-17/TNF-α axis (Figure 3F–J; all *p* < 0.05) when compared with the control group. Even the relatively low-potency DXM group exhibited a similar ability to inhibit the protein expression of the IL-23/IL-17/TNF-α axis in psoriasis-like skin when compared with high-potency steroids such as ESP and CLO (Figure 3F–J).

### 3.5. In the Crossover Study, 0.05% DXM Demonstrated Superior Therapeutic Effects to Esperson

To validate the potential therapeutic effects of the relatively low dose of DXM present in the formulated product, a crossover study was conducted using ESP as a comparator. The study design, illustrated in Figure 4A, involved three different treatment groups: (1) a switch from DXM treatment to no treatment (DXM to NT); (2) a switch from DXM treatment to Esperson treatment (DXM to ESP); (3) a switch from Esperson treatment to DXM treatment (ESP to DXM). Following the initial treatment phase on Day 6, DXM was observed to be a more effective treatment option than ESP in terms of morphology (Figure 4B), histology (Figure 4C), and barrier function (Figure 4D–F), aligning with the results obtained from the comparison study. After the crossover treatment phase on Day 15, both the DXM to ESP and ESP to DXM treatment groups displayed a normal untreated-like skin morphology (Figure 4B) and histology (Figure 4C) pattern. The group that switched from DXM to NT treatment exhibited a miniature skin-scale morphology (Figure 4B) and a thicker epidermis, with 4–5 layers of keratinocytes (Figure 4C). The group that switched from ESP to DXM demonstrated a significant reduction in transepidermal water loss (TEWL) (*p* < 0.01; Figure 4D) and skin erythema (*p* < 0.05; Figure 4F) values. It also experienced a notable increase in skin hydration on Day 15 compared with Day 6 (*p* < 0.01; Figure 4E). These findings suggest that the DXM-loaded formulation could effectively improve the therapeutic effect in psoriasis.

### 3.6. The DXM-Loaded Formulation Represented a Safe Topical Formulation

The comparison study indicated that DXM could serve as a promising alternative to treating psoriasis. Subsequently, a skin irritation study was conducted to evaluate the safety of the formulation loaded with DXM. Following six consecutive days of application, the morphology (Figure 5A), histology (Figure 5B), and barrier function values (Figure 5C–E) of the treated group were compared with those of a normal untreated group. No significant differences were observed, and there was no detection of desoximetasone in blood samples collected after a 6-day application period. These findings suggest that the DXM-loaded formulation represents a safe topical treatment option.

### 3.7. Skin Penetration Parameters and Deposition Amounts

Skin penetration and deposition studies were conducted to elucidate the differences in dermato-pharmacokinetics between DXM and ESP. Table 1 provides an overview of the penetration parameters of desoximetasone in DXM (0.05% desoximetasone loaded into a designed formulation) and the Esperson ointment (0.25% desoximetasone) when applied to normal murine skin for a duration of 6 h. The results demonstrated that ESP exhibited a skin flux rate that was two times faster than that of DXM (*p* < 0.05; Student’s *t*-test), along with a cumulative amount that was also two times higher than that of DXM (*p* < 0.05; Student’s *t*-test). Conversely, DXM demonstrated a shorter lag time compared with ESP in achieving a steady-state skin flux (*p* < 0.05; Student’s *t*-test). There was no significant difference observed in the permeability coefficient (Kp) between the two formulations (*p* = 0.067).

Figure 5F illustrates the skin deposition amounts of DXM for both formulations. The AUC_0–6_ (area under the curve from 0 to 6 h) of ESP (886.24 ± 62.43 μg/g tissue·h) was nearly five times higher than that of DXM (182.76 ± 51.53 μg/g tissue·h; *p* < 0.01; Student’s *t*-test). These findings indicate that ESP exhibited faster skin penetration, a higher cumulative amount, and a greater skin deposition amount compared with DXM.

## 4. Discussion

In this study, we evaluated the inhibitory effects of a designed formulation on CYP3A4 protein expression and its therapeutic potential for psoriasis-like skin lesions. The results showed that the formulated product demonstrated a superior inhibitory effect on CYP3A4 protein expression, as expected. The 0.05% desoximetasone treatment was selected; we observed it to be a more effective treatment option than Esperson and clobetasol for both the barrier function and morphology in a psoriasis-like skin lesion model. The 0.05% desoximetasone treatment—which is a moderate-potency steroid, typically used for children [20]—demonstrated a similar potential to inhibit the protein expression of the IL-23/IL-17/TNF-α axis in psoriasis-like skin when loaded into the designed formulation compared with high- to super-potency steroids, including Esperson and clobetasol [20]. The crossover study confirmed the potential therapeutic effect of the 0.05% desoximetasone-loaded formulation; it represented a safe topical formulation. The findings suggest that the developed formulation of desoximetasone could be an effective and rapid treatment option for psoriasis-like skin lesions, providing a potential alternative to high-potency steroids.

Our microemulsion formulation was compared with a commercially available moisturizing cream that contained the same percentage of CYP3A4 inhibitors. The immunohistochemical staining results demonstrated that treatment with the microemulsion formulation effectively inhibited the CYP3A4 protein expression from the upper layer to the lower layer of keratinocytes. The treatment with the cream formulation only inhibited the upper layer. The findings implied that the CYP3A4-inhibitory formulation components had to be delivered into the deep epidermis for effective CYP3A4 protein inhibition. The nature of the topical delivery system could impact the skin penetration of drugs, affecting their delivery to the target site [21,22]. Microemulsions and creams are two types of topical delivery systems that differ in their physicochemical properties, affecting drug penetration and delivery. By adequately selecting penetration promoters for microemulsion components and optimizing the structure and viscosity of the microemulsion, the drug release kinetics and the degree of skin penetration can be controlled [22]. However, the skin barrier can limit drug penetration and hinder delivery to the intended site. A cream-based delivery system must possess appropriate physicochemical properties to overcome challenging skin barriers presented by a densely packed stratum corneum and tight junctions between keratinocytes [23,24,25]. CYP3A4 activity may also be a factor affecting drug metabolism [26]. To assess the protein activity inhibition ability, a CYP3A4 activity assay kit (Abcam) was selected, but CYP3A4 activity was not detected in two independent samples. This may have been due to the commercially available activity assay kits being designed for liver samples, not epidermis samples. Smith et al. detected skin CYP3A4 enzyme activity using a benzyl-*O*-methyl-cyanocoumarin substrate model [27]. Thus, a suitable sensitive assay method or kit is required to confirm the presence or absence of CYP3A4 activity in the epidermis.

Over the past few years, the treatment of psoriasis has undergone significant advances. For patients with moderate to severe plaque psoriasis, biologics that inhibit TNF-α, p40IL-12/23, IL-17, and p19IL-23, as well as an oral phosphodiesterase 4 inhibitor, have emerged as promising therapies [28]. For individuals with mild to moderate psoriasis, traditional topical treatments such as corticosteroids and vitamin D analogs continue to be first-line therapy [29]. In preliminary studies, the topical treatment product Daivobet (50 μg calcipotriol and 0.5 mg betamethasone ointment; LEO Laboratories Limited, Ireland) was selected for comparison. A recent study [30] reported no changes in calcium homeostasis or the hypothalamic–pituitary–adrenal axis in humans after Daivobet treatment. Our animal study observed that it caused a low body temperature and weight loss, leading to the death of three mice. It was hypothesized that the calcipotriol component of Daivobet induced hypercalcemia, resulting in a significant increase in blood calcium levels and severe adverse effects, precluding its use in psoriasis animal treatment studies.

Desoximetasone has therapeutic potential in the treatment of skin inflammation and autoimmune disorders [20]. In the present study, we evaluated the dermato-pharmacokinetics of desoximetasone loaded into a designed microemulsion (DXM) and a commercial Esperson cream (ESP) with their therapeutic efficacy in a mouse model of psoriasis. Our findings indicated that DXM was more effective than ESP at improving dermatitis and reducing the recurrence of psoriasis. However, ESP exhibited faster skin penetration and a higher cumulative deposition compared with DXM, possibly due to the high loading amount of desoximetasone (0.25%). The lack of a significant difference in the permeability coefficient (Kp) between the two formulations suggested that the molecular structures and physicochemical properties of the active compound—desoximetasone—contributed to the observed differences in the skin flux and deposition. This finding aligned with previous studies that demonstrated that the physicochemical properties of drugs play a crucial role in their skin permeation and deposition [21,22]. In contrast, we observed that CYP3A4 inhibition could result in an increased therapeutic effect and reduced loaded amount of active compound, indicating that metabolism by this enzyme could play a significant role in the pharmacokinetics of DXM. Our results suggest that inhibiting the metabolism of active compounds should be considered in the development and optimization of topical formulations for dermatological diseases.

## 5. Conclusions

In this study, we demonstrated that the excipient used in the microemulsions had the ability to suppress the expression of the CYP3A4 protein in a mouse epidermis. The formulated microemulsion effectively treated psoriasis-like skin lesions with a lower dose of desoximetasone—representing a low-potency steroid—surpassing the effectiveness of high-potency steroids such as Esperson and clobetasol. A crossover study confirmed the potential therapeutic benefits of the microemulsion; these may also help reduce psoriasis recurrence. Overall, this research highlights the potential of our microemulsion as a safe and effective topical treatment option for psoriasis-like skin lesions. Inhibiting the metabolism of active compounds should be considered when developing and optimizing topical formulations for dermatological diseases to increase the therapeutic effects.

## Figures and Tables

**Figure 1 pharmaceutics-15-02016-f001:**
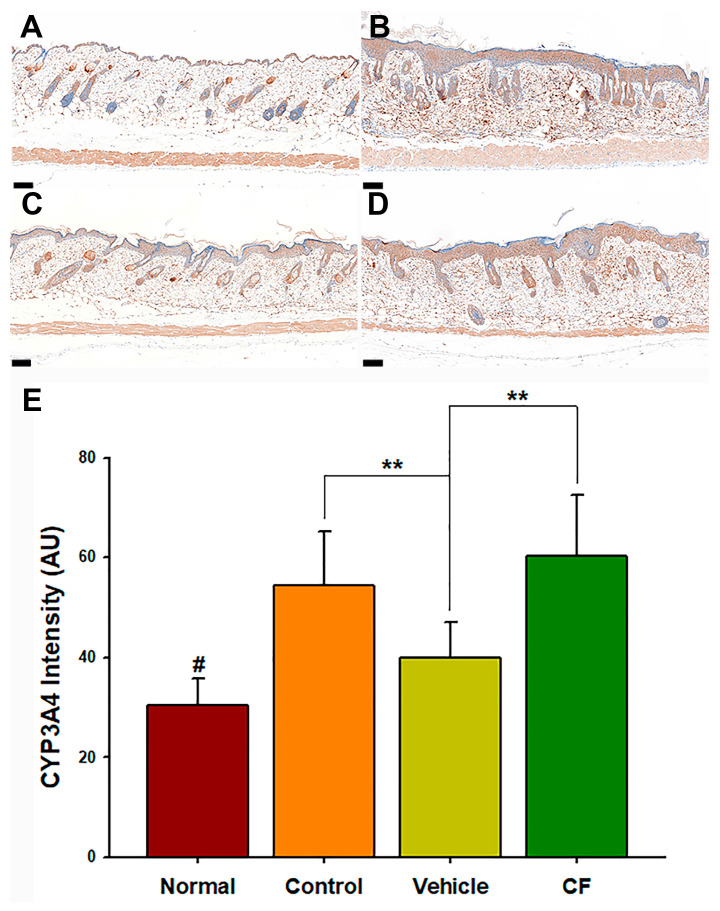
Unique inhibitory effect of the designed formulation on CYP3A4 protein expression. Results of immunohistochemical staining for CYP3A4 protein expression in the epidermis: (**A**) normal (healthy untreated skin); (**B**) control (IMQ-induced psoriasis-like skin); (**C**) vehicle (designed formulation for the treatment of IMQ-induced psoriasis-like skin); (**D**) CF (commercially available moisturizing cream with the same percentage of components as the CYP3A4-inhibited formulation used to treat psoriasis-like skin). (**E**) The counting results of immuno-intensity indicated that the designed formulation reduced the expression of CYP3A4 protein in keratinocytes after 6 days of application on IMQ-induced psoriasis-like mice. Scale bar: 100 μm. Statistical analysis: # *p* < 0.05 compared with all other treatment groups; ** *p* < 0.01 (one-way ANOVA; post hoc Scheffe); mean ± SD; *n* = 6.

**Figure 2 pharmaceutics-15-02016-f002:**
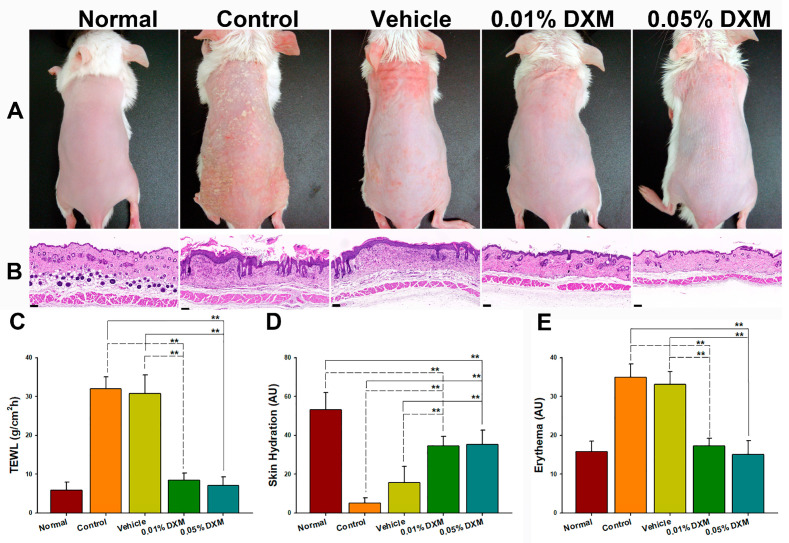
Effects of low-dose DXM loaded with the designed formulation on morphological changes and skin barrier restoration: (**A**) morphology; (**B**) hematoxylin and eosin (H&E) staining; (**C**) transepidermal water loss (TEWL); (**D**) skin hydration; (**E**) skin erythema. Scale bar: 100 μm; DXM: desoximetasone. Statistical analysis: ** *p* < 0.01 (one-way ANOVA; post hoc Scheffe); mean ± SD; *n* = 6.

**Figure 3 pharmaceutics-15-02016-f003:**
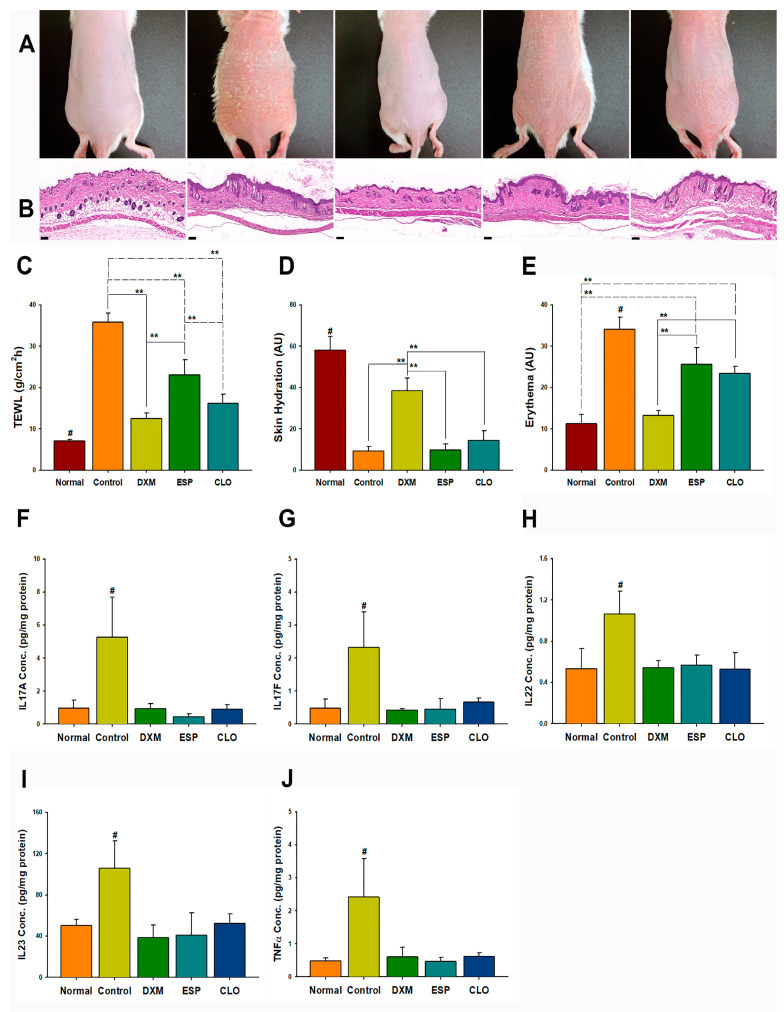
Comparative therapeutic efficacy of DXM, Esperson, and clobetasol in inhibiting IL-23/IL-17/TNF-α-axis protein expressions in psoriasis-like skin: (**A**) morphology; (**B**) H&E staining; (**C**) TEWL; (**D**) skin hydration; (**E**) skin erythema; (**F**) IL-17A, (**G**) IL-17F, (**H**) IL-22, (**I**) IL-23, and (**J**) TNF-α protein expressions. Scale bar: 100 μm; DXM: 0.05% desoximetasone loaded into the designed formulation; ESP: Esperson (0.25% desoximetasone ointment; Sanofi, Handok Inc, Seoul, Korea); CLO: clobetasol (0.5 mg/gm clobetasol ointment; Sinphar Pharmaceutical Co. Ltd., Taipei, Taiwan). Statistical analysis: # *p* < 0.05 compared with all other treatment groups; ** *p* < 0.01 (one-way ANOVA; post hoc Scheffe); mean ± SD; *n* = 6.

**Figure 4 pharmaceutics-15-02016-f004:**
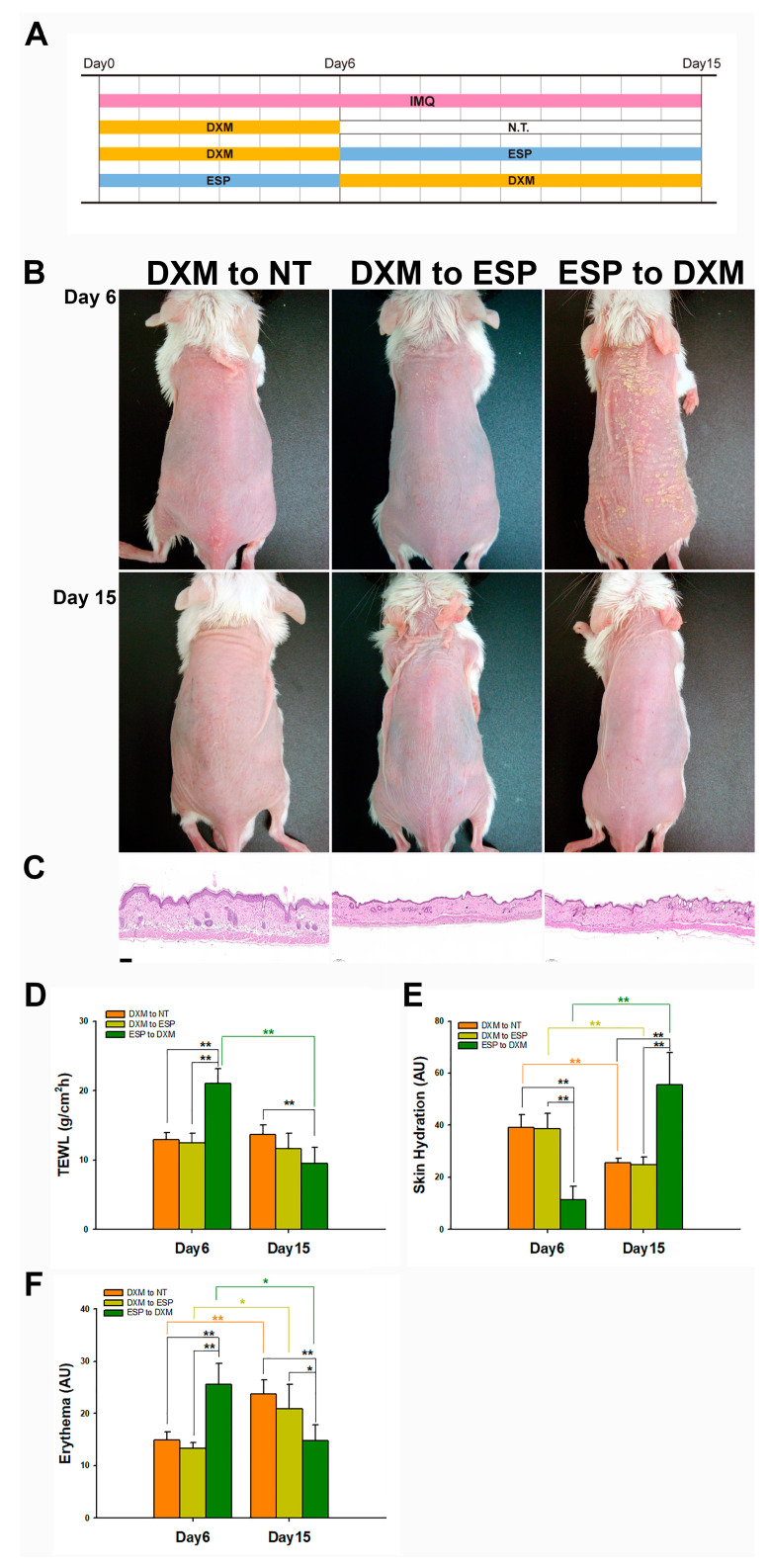
Comparative therapeutic efficacy of 0.05% DXM and Esperson in a crossover study. In the crossover study, 0.05% DXM demonstrated superior therapeutic effects to Esperson. The ESP to DXM treatment group exhibited a normal skin morphology and a thinner epidermis on Day 15 compared with the DXM to ESP and DXM to NT treatment groups: (**A**) study design; (**B**) morphology; (**C**) H&E staining; (**D**) TEWL; (**E**) skin hydration; (**F**) skin erythema. Scale bar: 100 μm; DXM: 0.05% desoximetasone loaded into the designed formulation; NT: no treatment; ESP: Esperson (0.25% desoximetasone ointment; Sanofi, Handok Inc., Seoul, Korea). Statistical analysis: * *p* < 0.05; ** *p* < 0.01 (one-way ANOVA, post hoc Scheffe); mean ± SD; *n* = 4.

**Figure 5 pharmaceutics-15-02016-f005:**
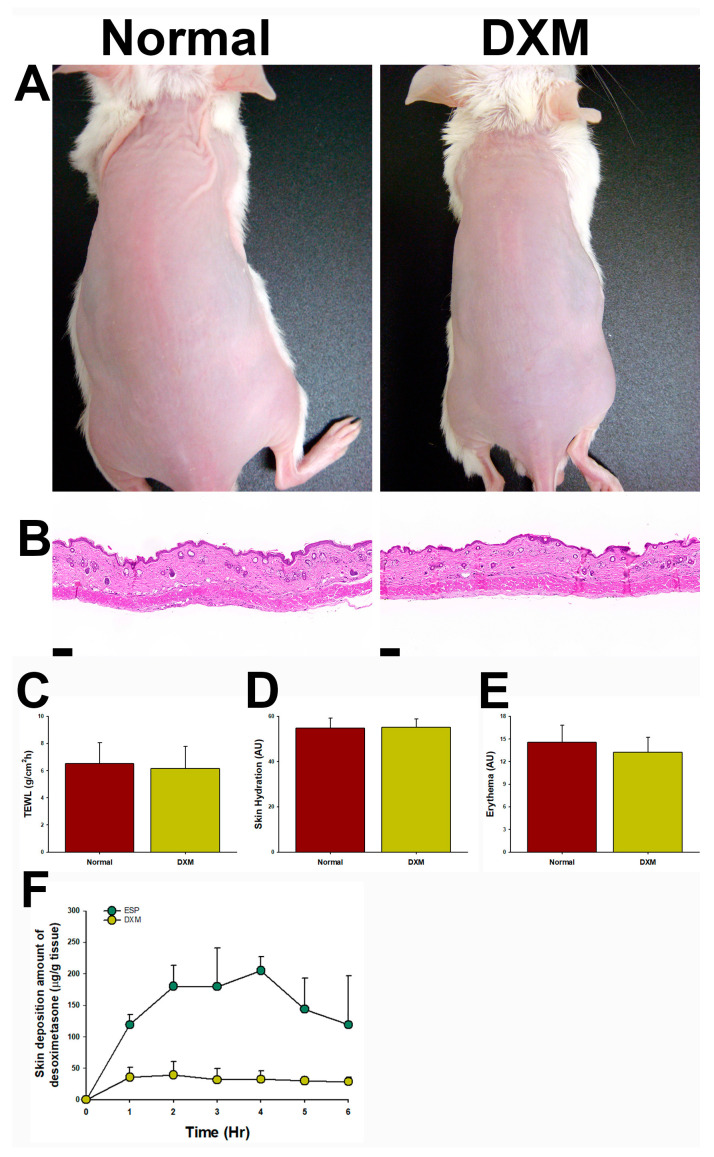
Safety and skin deposition of DXM-loaded formulation. The DXM-loaded formulation represented a safe topical formulation, supported by (**A**) morphology, (**B**) H&E staining, (**C**) TEWL, (**D**) skin hydration, and (**E**) skin erythema assessments. Scale bar: 100 μm. After six consecutive days of application, there were no differences observed between the DXM-loaded formulation treatment and the normal untreated group. (**F**) Skin deposition amounts of desoximetasone. DXM: 0.05% desoximetasone loaded into the designed formulation; ESP: Esperson (0.25% desoximetasone ointment; Sanofi, Handok Inc., Seoul, Korea). Student’s *t*-test; mean ± SD; *n* = 4.

**Table 1 pharmaceutics-15-02016-t001:** Penetration parameters of desoximetasone in the designed formulation and Esperson ointment through normal murine skin after 6 h.

Parameter (Unit)	DXM	ESP
Js (ng/cm^2^·h)	1.46 ± 0.56	3.05 ± 0.90 *
tlag (h)	1.24 ± 0.24 *	1.57 ± 0.11
Q (ng/cm^2^)	6.89 ± 2.66	13.88 ± 4.71 *
Kp (×10^−3^ cm/h)	14.63 ± 5.64	7.95 ± 2.05

DXM: 0.05% desoximetasone loaded into the designed formulation; ESP: Esperson (0.25% desoximetasone ointment; Sanofi, Handok Inc., Korea); Js: steady-state flux; tlag: lag time; Q: cumulative amount, where Q = concentration × volume of diffusion cell/area; Kp: permeability coefficient, where Kp = Js/drug concentration in donor cells. Statistical analysis: * *p* < 0.05; Student’s *t*-test; mean ± SD; *n* = 3.

## Data Availability

The data that support the findings of this study are available upon request to the corresponding authors.

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
