# Peer review of "A Promising Approach to Treat Psoriasis: Inhibiting Cytochrome P450 3A4 Metabolism to Enhance Desoximetasone Therapy"

_pharmaceutics, 2023, doi:10.3390/pharmaceutics15082016_

Round 1

Reviewer 1 Report

Thank you for submitting your manuscript on improvement of topical treatment of psoriasis.

- The title should be changed. "to treating psoriasis" - to treat

- p.1, l. 18: "treat skin conditions such as dermatoses, skin allergies, and psoriasis". Psoriasis and skin allergies are dermatoses.

- p.1, l. 39 - the same.

- Methods: Please mention the number of mice treated in the various groups.

- p.7, l.263: "DXM showed the more effective therapeutical effect than Esperson and Clobetasol" - DMX was more effective than....

- p.9, l.289: "3.4. DXM, ESP, and CLO Inhibit IL-23/IL-17/TNF-α axis protein..."

   - inhibit.

- p.10, l.319: "improve dermatitis and may help reduce the recurrence of psoriasis" - psoriasis has been investigated. Please avoid the nonspecific term dermatitis. An effect on psoriasis recurrence has not been demonstrated by your experiment.

- p.14, l. 431: "It has been reported that the therapeutic potential of desoximetasone is in treating skin inflammation and autoimmune disorders" - 

Desoximetahsone has a therapeutic potential in the treatment of ...

General aspects: Please explain why desoximethasone was used instead of any other corticosteroid? 

The paper needs significant improvement in English.

Author Response

Reviewer 1

Thank you for submitting your manuscript on improvement of topical treatment of psoriasis.

Response: We thank the reviewer for the valuable suggestions and recommendations regarding this research. 

- The title should be changed. "to treating psoriasis" - to treat

Response: We have revised the paper title as “A promising approach to treat psoriasis: inhibiting cyto-chrome P450 3A4 metabolism to enhance desoximetasone therapy”.

- p.1, l. 18: "treat skin conditions such as dermatoses, skin allergies, and psoriasis". Psoriasis and skin allergies are dermatoses.

- p.1, l. 39 - the same.

Response: We have revised both the sentences as “Desoximetasone, a topical corticosteroid and cytochromes P450 3A4 substrate, is used to treat skin conditions such as skin allergies, atopic dermatitis, and psoriasis. (page 1, Line 18-19, and 39-40).

- Methods: Please mention the number of mice treated in the various groups.

Response: We have revised and provided the number of mice treated in the various groups.

- p.7, l.263: "DXM showed the more effective therapeutical effect than Esperson and Clobetasol" - DMX was more effective than....

Response: We have revised the sentences as “DMX was more effective than Esperson and Clobetasol (page 7, Line 280-281).

- p.9, l.289: "3.4. DXM, ESP, and CLO Inhibit IL-23/IL-17/TNF-α axis protein..."

   - inhibit.

Response: We have revised the sentences as “DXM, ESP, and CLO inhibit IL-23/IL-17/TNF-α axis protein in psoriasis-like skin” (page 9, Line 308-309).

- p.10, l.319: "improve dermatitis and may help reduce the recurrence of psoriasis" - psoriasis has been investigated. Please avoid the nonspecific term dermatitis. An effect on psoriasis recurrence has not been demonstrated by your experiment.

Response: We have revised the sentences as “These findings suggest that the DXM-loaded formulation can effectively improve the therapeutic effect in psoriasis.” (page 10, Line 340-341).

- p.14, l. 431: "It has been reported that the therapeutic potential of desoximetasone is in treating skin inflammation and autoimmune disorders" -

Desoximetahsone has a therapeutic potential in the treatment of ...

Response: We have revised both the sentences as “Desoximetahsone has a therapeutic potential in the treatment of skin inflammation and autoimmune disorders.” (page 14, Line 458-459).

General aspects: Please explain why desoximethasone was used instead of any other corticosteroid?

Response: Desoximetasone (C22H29FO4; , MW  = 376.5),  which is a substrate of CYP3A4, is a potent topical corticosteroid used to alleviate various skin conditions, such as skin allergies, atopic dermatitis, and psoriasis. Desoximetasone is also safe with topical bioavailability steroids; it is used as a routine topical steroid at several medical centers in Taiwan, including Cathay General Hospital.

Comments on the Quality of English Language

The paper needs significant improvement in English.

Response: This article has been professionally edited by the MDPI English Editing Center's Specialist service.

Reviewer 2 Report

This is a very good manuscript on an important topic. Very well-designed study, with very well-presented results. The study is clearly justified and executed without any flaws. I recommend to accept the study as it is.

Author Response

Reviewer 2

This is a very good manuscript on an important topic. Very well-designed study, with very well-presented results. The study is clearly justified and executed without any flaws. I recommend to accept the study as it is.

Response: We sincerely appreciate the approval and recommendation from the reviewer for the recognition of our research findings.
